# Animal Venom for Medical Usage in Pharmacopuncture in Korean Medicine: Current Status and Clinical Implication

**DOI:** 10.3390/toxins13020105

**Published:** 2021-02-01

**Authors:** Soo-Hyun Sung, Ji-Won Kim, Ji-Eun Han, Byung-Cheul Shin, Jang-Kyung Park, Gihyun Lee

**Affiliations:** 1Department of Policy Development, National Development Institute of Korean Medicine, Seoul 04554, Korea; koyote10010@nikom.or.kr (S.-H.S.); cactus@nikom.or.kr (J.-W.K.); jieun2342@nikom.or.kr (J.-E.H.); 2Division of Clinical Medicine, School of Korean Medicine, Pusan National University, Yangsan 50612, Korea; drshinbc@pusan.ac.kr (B.-C.S.); vivat314@pusan.ac.kr (J.-K.P.); 3College of Korean Medicine, Dongshin University, Naju 58245, Korea

**Keywords:** animal venom, bee venom, snake venom, toad venom, pharmacopuncture

## Abstract

Animal venoms, widespread throughout the world, are complex mixtures, the composition of which depends on the venom-producing species. The objective of this study was to contribute to the development of animal venom-based medicines by investigating the use of animal venom pharmacopuncture in Korean medicine (KM) institutions. We surveyed 256 public health centers from 1 through 31 October 2019 as guided by the Ministry of Health and Welfare (MoHW). A questionnaire developed by an expert group was distributed and collected for statistical analysis. The survey identified three types of animal venom-based pharmacopuncture: bee, snake, and toad venoms. The medications are based on a single animal venom ingredient and produced in 11 external herbal dispensaries (EHDs). Each animal venom is processed, refined, and freeze-dried in a cleanroom to produce a powder formulation that is later measured, diluted, filtered, filled, sealed, sterilized, and packaged as pharmacopuncture injections used in KM institutions. Bee venom therapy is effective in treating musculoskeletal pain, snake venom therapy is effective in controlling bleeding during surgery, and toad venom therapy is effective in cancer treatment. The study suggests that bee, snake, and toad venoms could be used in medical institutions and have the potential for drug development.

## 1. Introduction

Animal venoms, widespread worldwide, are composed of a variety of proteins and peptides that were developed through millions of years of evolution [1,2,3]. They are complex mixtures that vary depending on the venom-producing species [2]. The most known and studied venoms originate from bees, snakes, scorpions, and spiders [1,2].

Venomous animals are a valuable resource for the development of therapeutics [4]. Bee venom therapy is used to treat musculoskeletal diseases (e.g., low back pain, knee osteoarthritis), Parkinson’s disease, adhesive capsulitis, and polycystic ovary syndrome [5]; snake venom-based drugs are used to treat hypertension, heart attack, acute coronary syndrome, stroke, pulmonary embolism, and other diseases [6]. While animal venoms are toxic, they can have therapeutic effects [7].

Korean medicine (KM) institutions have been using animal venoms in their pharmacopuncture therapies: animal venoms are injected into acupunctural points using syringes [8]. Pharmacopuncture, also known as herbal acupuncture, is a new form of acupuncture treatment that injects herbal medicine into acupuncture points [9]. In South Korea, pharmacopuncture treatment is one of the most commonly used methods in traditional Korean medicine clinics [10]. Official insurance claims statistics for pharmacopuncture are only available for automobile insurance coverage; it has been reported that approximately 1,555,000 pharmacopuncture sessions were administered to 168,089 patients for related treatments in 2014 [11].

Chinese pharmaceutical companies produce acupunctural medicines approved by the food and drug authorities [12]; these medicinal materials are used in Chinese medicine institutions to treat patients. In South Korea, such medicinal products are made in external herbal dispensaries (EHDs) that employ the level of good manufacturing practice (GMP) required for injections [13]. EHDs are auxiliary facilities of KM institutions that prepare and supply various types of medicines (e.g., capsules, tablets, powders, pharmacopuncture, and decoction) to KM institutions [14].

The purpose of this study was to review the current use of animal venom pharmacopuncture medicines in KM institutions; how such medicines are prepared; for what disease treatment they are being used; and further, to provide basic data for the progressive, knowledge-based development of animal venom treatments derived from the understanding of the current status.

## 2. Results

### 2.1. Demographic Characteristics of Pharmacopuncture-EHD (P-EHD)

A total of 11 Pharmacopuncture-EHD (P-EHD) questions were included in this study. The characteristics of the respondents in this survey are shown in Appendix A. The average number of pharmacopuncture medicines produced in the 11 P-EHDs was average 18.4.

### 2.2. Preparation Status of Pharmacopuncture

A total of 1,851,896 vials of 46 different pharmacopuncture medicines were produced in 11 P-EHDs in 2018 alone. The status of the top 20 frequently prepared pharmacopuncture medicines produced in P-EHDs and their ingredients are shown in Table 1.

### 2.3. The Preparation Process of Animal Venom Pharmacopuncture

Bee venom is extracted from the venom pouch of worker bees (*Apis mellifera, ligustica*) [15]. Apply a weak current of a low voltage, 3–6V, to a living bee. Then, take the venom coming out of the stinger [15]. Snake venom is extracted from the venom gland of *Agkistrodon halys Pallas* [15]. Hold a snake by its head and neck. Then, place a vessel under its fangs and press the head where the venom gland is located gently to press the venom out of it. This process is called “snake milking” [15,16]. Toad (*Bufo bufo gargarizans Canto*) venom is extracted from the skin and parotid glands [15]. The production process of the injectable pharmacopuncture is the same for all three venoms (Figure 2). Venoms are processed, refined, and freeze-dried in a clean room into a powder formulation [15]. The venom powder goes through purification and dilution using distilled water in a specific concentration [15]. The venom is then filled, sealed, sterilized, and packaged as pharmacopuncture injections for clinical use (Figure 2).

### 2.4. Clinical Effectiveness of Animal Venom Therapy

Table 2 summarizes the clinical evidence for animal venom therapy use and efficacy [17,18,19,20,21,22,23,24,25,26,27,28,29,30,31,32,33,34,35,36,37,38,39,40,41,42,43,44,45,46,47,48,49,50,51,52,53,54,55,56,57,58,59,60,61]. Bee venom therapy is effective in musculoskeletal pain such as post-stroke shoulder pain [18,42,43,44,45], shoulder pain [19], chronic low back pain [23,24,25], neck pain [32], temporomandibular disorder [34], wrist sprain [35], and acute ankle sprain [37,38]. Snake venom therapy has been reported to be clinically effective in hemorrhage control in bi-maxillary orthognathic surgery [50], fracture-related hip hemiarthroplasty [51], and transvesical prostatic adenomectomy [54]. The clinical indications of toad venom therapy include gastric cancer [57], liver cancer [58], and metastatic bone tumors [59]. Eighteen studies [17,18,19,20,22,23,24,25,31,32,37,40,41,43,48,57,58,59] reported moderate to severe adverse events in the animal venom treatment groups.

## 3. Discussion

To our knowledge, this study is the first to report on the three animal venom therapies used in KM institutions in South Korea. It covers the current status of the production of pharmacopuncture medicines based on animal venoms, their production processes, and indications. The government played a leading role in this study, which aimed to understand the current status of the production of animal venom pharmacopuncture medicines in EHDs. All of the pharmacopuncture medicines used in KM institutions are being produced in EHDs; this study covered 11 of the 16 EHDs that produced pharmacopuncture medicines (68.8%). Since EHDs prepared these medicines following the prescriptions issued by KM institutions, the production records at EHDs can be considered to represent the current status of the pharmacopuncture medicines used in KM institutions. Therefore, this study provides an overall status of the use of animal venoms for therapeutic applications by KM institutions.

Bee venom products represent the second-largest quantity of pharmacopuncture medicines in South Korea, accounting for 99.06% of the total animal venom-based pharmacopuncture. While bee venom causes neurotoxic symptoms including local paresthesias, headache, dizziness, nausea and vomiting, muscle aches, and rarely, cerebrovascular infarcts [62,63,64,65], recent non-clinical studies demonstrated that bee venom has anti-inflammatory [66], anti-nociceptive [67], and anti-cancer activities [68]. The toxic reactions of bee venom therapy in clinical studies ranged from fatigue [69], erythematous plaques [70], pallor face [71], nausea [71], vomiting [71], and other minor side effects to more serious ones such as limb paralysis [72], dyspnea [72], unconsciousness [73], and death [73]. There have been many clinical and non-clinical studies reporting the side effects and benefits of bee venom therapy. However, due to the possibility of variation in the species of bees and the nutrients they take, the composition of bee venom can differ depending on geography. Therefore, it is necessary to standardize the material, evaluate its toxicity, and examine its benefits.

Snake venom is the 13th most frequently used pharmacopuncture medicine in KM institutions. While snake venoms can be cardiotoxic (e.g., arrhythmias, bradycardia, tachycardia and hypotension) [74], myotoxic (e.g., myalgias, myopathy and rhabdomyolysis) [74], neurotoxic (e.g., ptosis, external ophthalmoplegia, dysphagia, dysphonia and broken neck sign) [75], and nephrotoxic (e.g., ischemia and renal failure) [74], non-clinical studies of snake venom reported effects such as a decrease in blood pressure [76], regulation of blood coagulation [77], and anti-tumor [78] and analgesic activities [78]. The snake-venom-based medicines that have been approved by the Food and Drug Administration (FDA) are administered using tablets (captopril) or intravenous injections (eptifibatide and tirofiban) [79]. Further studies are needed to evaluate the efficacy and toxicity of the routes of administration (e.g., oral, muscle injection, or infusion, etc.).

Toad venom is the least frequently used animal venom in KM institutions (0.27%). While toad venom has toxic effects and causes gastrointestinal, mental, cardiac conduction, and arrhythmic disturbances [80], non-clinical studies have reported anti-inflammatory [81], antiplasmodial activity [82], and antiproliferative effects on cancer cells [83]. The treatment toxicity is the most important factor limiting its use [84]. Further studies should be carried out to evaluate the benefits and side effects of its use to treat inflammatory diseases.

As is the case with bee stings, animal venom was used as it was, in some cases. However, in most cases, it was used as diluted to an appropriate concentration for clinical purposes. Animal venoms may cause idiosyncratic reactions depending on the individual in addition to their toxic effect [85]. For this reason, it is recommended to have the patient take a certain amount before clinical applications, in order to determine whether the use or administration of such a substance causes adverse effects or not. Also, according to the analysis on the 1,192,667 cases of accidents caused by animal venom over 12 years in Brazil by Chippaux [86], snake bites were linked to the highest fatality rate of 0.43%, followed by bee stings, of which the fatality rate was 0.33%. Since animal venom has toxic effects, it is necessary to refine, dilute, or otherwise process it before clinical use.

This study has several limitations. First, this study was a literature review of the indications and benefits of pharmacopuncture medicines based on animal venoms, without any reviews on the clinical indications in the real world. Further surveys of KM doctors are needed to cover the clinical indications and side effects. Second, a literature review of indications and side effects used only articles published in South Korea, and only articles dealing with injectables were reviewed. International databases will need to be reviewed to evaluate animal venoms factors such as the species, extraction process, and administration routes. Third, this survey was conducted by the government, which measured its accuracy and trustworthiness. However, the validity of the questionnaire was not confirmed.

Despite these limitations, this is the first study reviewing the production volume and current status of animal venom-based pharmacopuncture medicines used in clinical practice in South Korea. Collecting further data is necessary for the standardization of pharmacopuncture medicine and safety of dosages and clinical applications.

## 4. Conclusions

This study reports that bee, snake, and toad animal venoms are being used in medical institutions and have the potential for drug development. Based on the clinical evidence for animal venom summarized in this study, it is expected that the development of therapeutic agents for target diseases will be conducted.

## 5. Methods

### 5.1. Overview of the Study

A survey of community health centers across the country was conducted with the cooperation of the Ministry of Health and Welfare to understand the nationwide status of the production and use of animal venom medicinal products in the country.

For this purpose, a systematically developed questionnaire was distributed to participating organizations. The process of producing animal venom-based pharmacopuncture medicines was analyzed using textbooks, research papers, and data from the EHDs currently producing such medicines.

### 5.2. Study Sample

The study sample consisted of 256 community health centers across the country that were responsible for establishing and supervising EHDs in South Korea.

### 5.3. Questionnaire Development and Distribution

To develop the questionnaire, a group of five experts was formed (including two KM specialists with an average of 10+ years of clinical experience of pharmacopuncture (J.K.P., S.B.C.), one PhD with 10+ years of experience in the KM field (G.L.), one expert in the GMP, and one expert in hazard analysis critical control point (HACCP)). The draft of the questionnaire was developed based on previous studies [87,88,89]. The final version of the questionnaire was agreed upon through a review and revision process (Appendix A).

To ensure nationwide representation, the questionnaire was sent together with official letters to 256 community health centers in South Korea through the Ministry of Health and Welfare over the period from 1 through 31 October 2019. A community health center has the authority to approve the establishment and supervision of an EHD. The centers were asked to complete the questionnaire based on the status in 2018; the community health centers were also instructed to complete the questionnaire only when their EHD produced pharmacopuncture medicines. To enhance the accuracy of the survey, it was requested that the questionnaire be completed by the KM pharmacist or the head of the KM institution that established the EHD.

### 5.4. Questionnaire Items

To obtain accurate information on the use of animal venoms in pharmacopuncture, the survey items consisted of two items concerning the basic status of EHDs (i.e., the type of the medical institution that set up the EHD and the location) and three items relating the production of pharmacopuncture medicines (the type of pharmacopuncture, herbal ingredients, and production volume). The questionnaire was designed so that the five items could be completed in a Microsoft Office Excel file (verson 16.0, 2013, Microsoft^®^, Redmond, WA, USA).

### 5.5. Data Collection

Completed questionnaires that were sent back to us from the Ministry of Health and Welfare were reviewed for the integrity of the completed data in Excel files and sorted out by the relevant items. For any missing or incomplete data, the EHD was called or contacted via email to request additional or missing data and correct the incorrect data as much as possible.

### 5.6. Data Analysis

The data gathered using the questionnaire were processed using SPSS version 21.0 (IBM, Armonk, NY, USA). An explorative data analysis (EDA) was conducted. A summary of the data was prepared including the average, standard deviations, and frequency (*n*, %). Any incomplete data were treated as missing.

### 5.7. Literature Search of the Production Process and Clinical Effectiveness of Animal Venom Medications

The survey identified 46 types of pharmacopuncture medicines; three animal venoms (bee, snake, and toad venoms) were selected by the reviewing experts. The external experts were a professor of KM university who is a pharmaceutical board member at the national hospital, a KM professor who majored in herbal medicines, a general manager of an EHD with 8+ years of experience in preparing pharmacopuncture, and a quality control (QC) team leader from a pharmaceutical company that manufactures herbal drugs.

The researcher also reviewed textbooks used in South Korean medical schools for pharmacopuncture and related academic papers to review the production process and indications of the three selected animal venoms. The following electronic databases using keywords “bee venom”, “snake venom”, and “toad venom” were searched to identify relevant studies for inclusion in the review from inception to September 2020: Pubmed, EMBASE, Medline, Cochrane library, and six Korean databases (National Digital Science Library (NDSL), the Korean Traditional Knowledge Portal, KoreaMed, the Oriental Medicine Advanced Searching Integrated System (OASIS), the Research Information Sharing Service (RISS), and The National Library of Korea). All identified clinical studies and systematic reviews were examined to understand the clinical indications and effects of animal venom-based therapy. Two researchers (S.S.H. and J.E.H.) independently reviewed and screened the titles and abstracts of the retrieved studies based on predefined eligibility criteria. Two independent reviewers (J.W.K. and B.C.S.) extracted the treated diseases and the effectiveness of animal venom-based therapy. The systematic review reflected the results of the meta-analysis.

## Figures and Tables

**Figure 1 toxins-13-00105-f001:**
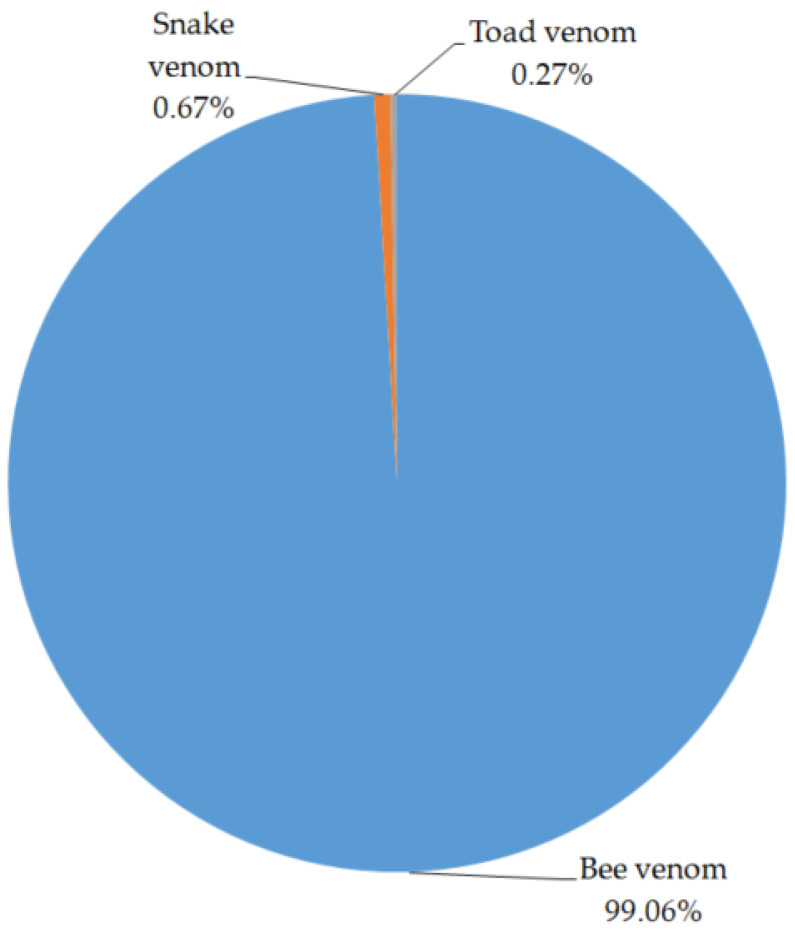
Preparation status of animal venom pharmacopuncture.

**Figure 2 toxins-13-00105-f002:**
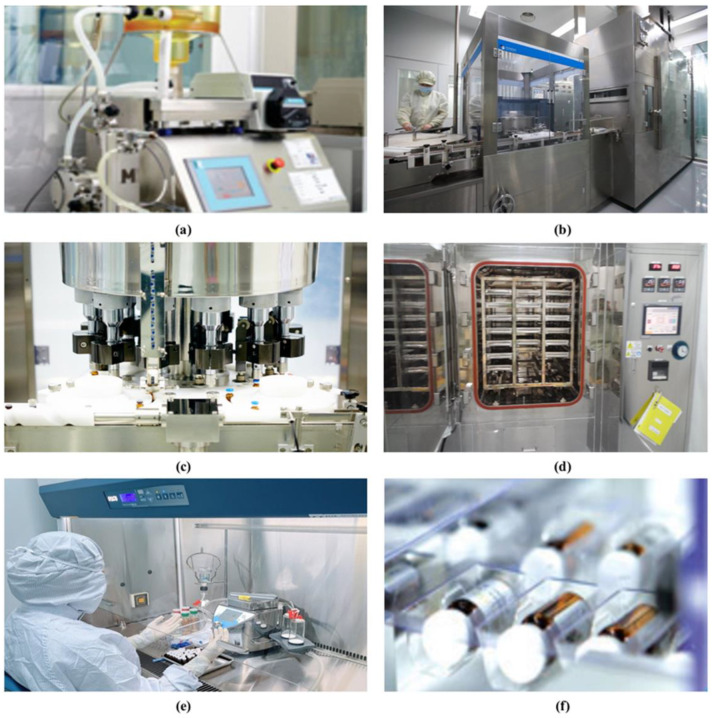
Preparation process of animal venom pharmacopuncture. Animal venom is prepared with pharmacopuncture through the following process; (**a**) venom is filted for removing impurities and diluted with distilled water (**b**) vials are cleaned and sterilized (**c**) venom is filled and sealed into each vial (**d**) prepared pharmacopuncture is sterilized (**e**) quality inspections (e.g., sterility test, endotosin test, and insoluble particulate matter test and pH test) are conducted and (**f**) paking pharmacopuncture for clinical use.

**Table 1 toxins-13-00105-t001:** Status and composition of pharmacopuncture.

NO.	Pharmacopuncture	Composition of Pharmacopuncture	Amount of Preparation (vial)	Ratio of Preparation
1	Jungseong-eohyeol	Salviae Radix, Persicae Semen, Commiphora myrrha Engler, Caesalpinia sappan Linné, Olibanum, Paeoniae Radix, Corydalis yanhusuo W.T.Wang, Gardeniae Fructus	1,373,686	23.45%
2	Bee venom	Bee venom	779,931	13.32%
3	Hwanglyeon-haedog	Coptidis Rhizoma, Phellodendri Cortex, Scutellariae Radix, Gardeniae Fructus	556,064	9.49%
4	Cheogchusin	Paeoniae Radix, Osterici seu Notopterygii Radix et Rhizoma, Araliae continentalis Radix, Cibotii Rhizoma, Eucommiae Cortex, Saposhnikoviae Radix, Acanthopanacis Cortex, Achyranthis Radix	490,990	8.38%
5	Haleupagopitumgeun	Harpagophyti Radix	272,330	4.65%
6	Jagyag-gamcho	Paeoniae Radix, Glycyrrhizae Radix et Rhizoma	153,530	2.62%
7	Hong-hwa	Carthami Flos, Carthami tinctorii Semen	146,380	2.50%
8	Jahageo	Hominis Placenta	115,909	1.98%
9	Wild ginseng	Wild ginseng	48,431	0.83%
10	Soyeom	Lonicerae japonicae Flos, Rehmanniae Radix, Forsythiae Fructus, Gardeniae Fructus	20,275	0.35%
11	Jug-yeom	Bamboo salt	8898	0.15%
12	Haeng-in	Armeniacae Semen	7528	0.13%
13	Snake venom	Snake venom	5291	0.09%
14	Cho-o	Aconiti Seoulense Tuber	4433	0.08%
15	Yang-geumhwa	Datura metel Linné	3530	0.06%
16	Chilpi	Rhus verniciflua Stokes	3000	0.05%
17	Cheong-ganbohyeol	Mori Folium	2600	0.04%
18	Toad venom	Toad venom	2114	0.04%
19	Miso	Salmon milt	2000	0.03%
20	Mahwang-cheono	Ephedrae Radix, Aconiti Tuber	1910	0.03%

The following quantities of animal venom preparations were made: bee venom (99.06%, *n* = 779,931), snake venom (0.67%, *n* = 5291), and toad venom (0.27%, *n* = 2114). All of these were composed of a single venom (Figure 1).

**Table 2 toxins-13-00105-t002:** Clinical evidence of animal venom therapy.

Type of Animal Venom	Treatment Diseases	Intervention (Concentration, Treatment Sessions, Amount of Venom Use)	Clinical Effects	Adverse Events	Type of Clinical Study
Bee venom	Musculoskeletal pain [17]	Bee venom acupuncture -concentration: 0.05–0.5 mg/mL -1 session: 0.01–1 mL -total 3–16 sessions: 0.05–30 mL	Positive	Skin hypersensitivity 2, itching 1, pain 2, pruritus 8, burning sensation 3	SR
Bee venom	Post-stroke shoulder pain [18]	Bee venom acupuncture -concentration: 0.1–0.5 mg/mL -1 session: 0.1–1.5 mL -total 6–12 sessions: 0.9–13.5 mL	Positive	Pain 2, pruritus 8, burning sensation 3	SR
Bee venom	Shoulder pain [19]	Bee venom acupuncture -concentration: 0.03–0.5 mg/mL -1 session: 0.1–1.5 mL -total 6–16 sessions: 0.6–14.8 mL	Positive	Pain 2, pruritus 8, burning sensation 3 Pruritus/local swelling/redness 30, mild, generalized swelling/aching 1	SR
Bee venom	Adhesive capsulitis [20]	(A) Bee venom acupuncture -concentration: 0.1 mg/mL -1 session: 0.4 mL (first visit), 0.6 mL (second visit), 0.8 mL (third visit), 1 mL (4–16 visit) -total 16 sessions: 14.8 mL (B) Bee venom acupuncture -concentration: 0.03 mg/mL -1 session: 0.4 mL (first visit), 0.6 mL (second visit), 0.8 mL (third visit), 1 mL (4–16 visit) -total 16 sessions: 14.8 mL	Positive	Pruritus/local swelling/redness 30, mild, generalized swelling/aching 1	RCT
Bee venom	Adhesive capsulitis [21]	(A) Bee venom acupuncture -concentration: 0.1 mg/mL -1 session: 0.4 mL (first visit), 0.6 mL (second visit), 0.8 mL (third visit), 1 mL (4–16 visit) -total 16 sessions: 14.8 mL (B) Bee venom acupuncture -concentration: 0.03 mg/mL -1 session: 0.4 mL (first visit), 0.6 mL (second visit), 0.8 mL (third visit), 1 mL (4–16 visit) -total 16 sessions: 14.8 mL	Positive	n.r.	RCT
Bee venom	Central post stroke pain [22]	Bee venom acupuncture -concentration: n.r. -1 session: 0.3 mL -total 6 sessions: 1.8 mL	Positive	No adverse events	RCT
Bee venom	Chronic low back pain [23]	Bee venom acupuncture -concentration: 0.05 mg/mL -1 session: 2 mL (first week), 4 mL (second week), 8 mL (third week) -total 6 sessions: 28 mL	Positive	Itching/sensation 4, headache 1, generalized myalgia 1	RCT
Bee venom	Chronic low back pain [24]	Bee venom acupuncture -concentration: 0.05 mg/mL -1 session: 0.6 mL -total 8 sessions: 4.8 mL	Positive	Itching 15, skin flare 5, edema 4, rash 2	RCT
Bee venom	Chronic low back pain [25]	Bee venom acupuncture -concentration: 0.3 mg/mL -1 session: 0.1 mL -total 5 sessions: 0.5 mL	Positive	Skin hypersensitivity 1	RCT
Bee venom	Delayed onset muscle soreness [26]	Bee venom gel -concentration: ultrasound gel and 1 mg/mL bee venom mixed at a ratio of 9:1 -1 session: n.r. -total 3 session: n.r.	Positive	n.r.	RCT
Bee venom	Herniated lumbar disc [27]	Bee venom acupuncture -concentration: 0.05–0.4 mg/mL -1 session: 0.01–0.15 mL -total 7 sessions: 0.07–1.05 mL	Positive	n.r.	RCT
Bee venom	Herniated lumbar disc [28]	Bee venom acupuncture -concentration: 0.25 mg/mL (early stage)–0.5 mg/ mL (last stage) -1 session: 0.25 mg/mL bee venom 0.1 mL, 0.5 mg/mL bee venom 1mL -total 7 sessions: 0.25 mg/mL bee venom 3 mL, 0.5 mg/mL bee venom 30 mL	Positive	n.r.	RCT
Bee venom	Knee osteoarthritis [29]	Bee venom acupuncture -concentration: 0.3 mg/mL -1 session: 0.5–1 mL -total 8 sessions: 4–8 mL	Positive	n.r.	RCT
Bee venom	knee osteoarthritis [30]	Bee venom acupuncture -concentration: bee venom powder 1 mg and 1 mL 0.5% lidocaine were mixed -1 session: 1.2 mL(1–3 weeks), 1.5 mL (4–12) -total 12 sessions: 17.1 mL	Positive	n.r.	RCT
Bee venom	Multiple sclerosis [31]	Bee venom sting therapy -concentration: bee sting -1 session: maximum 20 times -total 72 sessions: maximum 1440 times	Positive	Extreme swelling 2, itching 4, flulike symptoms 5, local tenderness/swelling/redness n.r.	RCT
Bee venom	Neck pain [32]	Bee venom acupuncture -concentration: 0.3 mg/mL -1 session: 0.1 mL -total: n.r.	Positive	Skin hypersensitivity 1	RCT
Bee venom	Pelvic inflammatory disease [33]	Bee venom gel -concentration: bee venom 20 μg/gel 1 g -1 session: n.r. -total 12 sessions: n.r.	Positive	n.r.	RCT
Bee venom	Temporomandibular disorder [34]	Bee venom ointment -concentration: 0.0005% -1 time: n.r. -total 42 times: n.r.	Positive	n.r.	RCT
Bee venom	Wrist sprain [35]	Bee venom acupuncture -concentration: 0.3 mg/mL -1 session: 0.05–10.5 mL -over 2 sessions: n.r.	Positive	n.r.	RCT
Bee venom	Polycystic ovary syndrome [36]	Bee venom gel -concentration: n.r. -1 session: 30–50 g -total 28 sessions: 840–1400 g	NS	n.r.	RCT
Bee venom	Acute ankle sprain [37]	Bee venom acupuncture -concentration: 0.3 mg/mL -1 session: 0.06 mL -total 7 sessions: 0.42 mL	Positive	Itching 1	RCT
Bee venom	Acute ankle sprain [38]	Bee venom acupuncture -concentration: 0.25 mg/mL, 0.1 mg/mL -1 session: 0.25 mg/mL 0.3 mL, 0.1 mg/mL 0.2 mL -total 3 sessions: 0.25 mg/mL 0.9 mL, 0.1 mg/mL 0.6 mL	NS	n.r.	RCT
Bee venom	Parkinson’s disease [39]	Bee venom acupuncture -concentration: 0.05 mg/mL -1 session: 1 mL -total 16 sessions: 16 mL	Positive	Itching 1	RCT
Bee venom	Parkinson’s disease [40]	Bee venom acupuncture -concentration: bee venom powder 1 mg and 20 mL normal saline were mixed -1 session: 1 mL -total 24 sessions: 24 mL	Positive	Mild pain/slight bleeding n.r., mild itching/mild swelling n.r.	RCT
Bee venom	Parkinson’s disease [41]	Bee venom acupuncture -concentration: bee venom 0.1 mg and 1 mL of NaCl 0.9% were mixed -1 session: 0.05 mL -total 11 sessions: 0.55 mL	NS	Redness/itching n.r., insomnia 1, nausea 3, fatigue 2, dyskinesia 1, bee venom specific IgE 18, bee venom specific IgG4 12	RCT
Bee venom	Post-stroke shoulder pain [42]	Bee venom acupuncture -concentration: n.r. -1 session: 0.3–0.6 mL -total 12 sessions: 3.6–7.2 mL	Positive	n.r.	RCT
Bee venom	Post stroke shoulder pain [43]	Bee venom acupuncture -concentration: 0.1 mg/mL -1 session: 0.6 mL -total 6 sessions: 3.6 mL	Positive	Pain 2, pruritus 8, burning sensation 3	RCT
Bee venom	Post-stroke shoulder pain [44]	Bee venom acupuncture -concentration: 0.5 mg/mL -1 session: 0.25–0.5 mL -total 12 sessions: 3–6 mL	Positive	n.r.	RCT
Bee venom	Post-stroke shoulder pain [45]	Bee venom acupuncture -concentration: 0.1–0.25 mg/mL -1 session: 0.1–1.5 mL -total 9 sessions: 0.9–13.5 mL	Positive	n.r.	RCT
Bee venom	Post-stroke shoulder pain [46]	Bee venom acupuncture -concentration: 0.05 mg/mL -1 session: 0.1 mL -total 6 sessions: 0.6 mL	NS	n.r.	RCT
Bee venom	Rheumatoid arthritis [47]	Bee venom acupuncture -concentration: 0.3 mg/mL -1 session: 0.2–1.0 mL -total 16 sessions: 3.2–16 mL	Positive	n.r.	RCT
Bee venom	Rheumatoid arthritis [48]	Bee venom sting therapy -concentration: bee sting -1 session: 5–15 times -total 24 sessions: 120–360 times	NS	Itching 60, redness 44, swelling 41, burning sensation 9, lymph node hypertrophy 2	RCT
Snake venom	Acute ischemic stroke [49]	Snake venom intravenous injection -concentration:0.5–1 U/kg -1 day: 0.5–1 U/kg then variable dose to keep fibrinogen level 70–130 mg/dL -total 7–14 days	Positive	n.r.	SR
Snake venom	Haemorrhage control in bi-maxillary orthognathic surgery [50]	Snake venom intravenous injection -concentration: 1 U/mL aqueous solution containing 0.9% sodium chloride, 0.3% phenol, and water for injection -1session: n.r. (single bolus dose before surgery)	Positive	No adverse events	RCT
Snake venom	Haemorrhage control in fracture-related hip hemiarthroplasty [51]	Snake venom intravenous injection -concentration: n.r. -1 session: 1 U -total 3 sessions: 3 U, before surgery (10–15 h preoperative) and repeated in equivalent volumes at 30 min preoperative and 12 h postoperative	Positive	No adverse events	RCT
Snake venom	Acute myocardial infarction [52]	Snake venom intravenous injection -concentration: 7.5 U/100mL glucose solution -1 session: 7.5 U snake venom was placed in a 100 mL 5% glucose solution, and reduced to 2.5 U -total 2–14 sessions	Positive	n.r.	RCT
Snake venom	Acute myocardial infarction [53]	Snake venom intracoronary or intravenous injection -concentration: n.r. -1 session: n.r. (single bolus dose before surgery)	Positive	n.r.	RCT
Snake venom	Haemostasis during transvesical prostatic adenomectomy [54]	Snake venom intravenous injection -concentration: n.r. -1session: 1 ample -total 14 sessions: 14 samples	Positive	n.r.	RCT
Snake venom	Acute cerebral infarction [55]	Snake venom intravenous injection -concentration: 0.0005 U/kg -1 day: 0.0005 U/kg × 2 times/day -total 7 days: 0.007 U/kg	Positive	No adverse events	RCT
Toad venom	Cancer-related pain [56]	Toad venom intravenous injection -concentration: n.r. -1 session: n.r. -total 1 week–30 days sessions: n.r.	Positive	n.r.	SR
Toad venom	Gastric cancer [57]	Toad venom intravenous injection -concentration: n.r. -1 session: 10–50 mL -total 28–112 days: n.r.	Positive	Leukopenia 113, nausea/vomiting 56, diarrhea 28	SR
Toad venom	Liver cancer [58]	Toad venom intravenous injection -concentration: n.r. -1 session: 10–30 mL -total sessions: n.r.	Positive	Nausea/vomiting 2	SR (network meta-analysis)
Toad venom	Metastatic bone tumors [59]	Toad venom capsule -concentration: n.r. -1 session: 500 mg -total 36 sessions: 18 g	Positive	Fever 3, nausea and vomiting 4, diarrhea 2, muscle soreness 3	RCT
Toad venom	Primary liver cancer [60]	Toad venom intravenous injection -concentration: n.r. -1 session: 20 mL toad venom was placed in a 250 mL 5% glucose solution -total 15 session: 300 mL (20 mL toad venom was placed in a 250 mL 5% glucose solution × 15 times)	Positive	n.r.	RCT
Toad venom	Advanced pancreatic adenocarcinomas [61]	Toad venom injection -concentration: -1session: 20 mL m^−2^ over 2 h -total 15 sessions: 300 mL m^−2^	NS	n.r.	RCT

n.r.: not reported; NS: not statistically significant effectiveness between groups; Positive: statistically significant effectiveness between groups; RCT: randomized controlled trials; SR: systematic review.

## Data Availability

The data will be made available upon reasonable request.

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
