# Peer review of "Animal Venom for Medical Usage in Pharmacopuncture in Korean Medicine: Current Status and Clinical Implication"

_toxins, 2021, doi:10.3390/toxins13020105_

Round 1

Reviewer 1 Report

Comments to Author:

The review " Animal venom as a medical usage in pharmacopuncture in Korean medicine: current status and clinical implication" describe use of animal venom (bee, snake and toad) pharmacopuncture in Korean medicine (KM) 256 public health centers institutions from December 2017 to January 2018.  Although this review to be interesting and innovative, there are some points that should be amended in order to improve the manuscript.

General comments:

The authors report “ The purpose of this study was to review the current use of animal-venom pharmacopuncture medicines in KM institutions, how such medicines are prepared, and for whatdisease treatment they are being used. Further, to provide basic data for the progressive, knowledge-based development of animal venom treatments derived from the understanding of the current status”.  However, in the present format, this review did not achieve these objectives. The authors should be more details about the toxic effect of these crude venom. Moreover, they could better describe the process of Preparation status of pharmacopuncture, especially the venom concentrations. The Questionnaire used in the study must be included in the review. In addition, reading the article I missed a more detailed table with the description of all the data obtained, highlighting the results of the therapy. Finally, the discussion can be improved based on clinical data.

Author Response

Reviewer 1

(Reviewer’s comment 1) The review " Animal venom as a medical usage in pharmacopuncture in Korean medicine: current status and clinical implication" describe use of animal venom (bee, snake and toad) pharmacopuncture in Korean medicine (KM) 256 public health centers institutions from December 2017 to January 2018. Although this review to be interesting and innovative, there are some points that should be amended in order to improve the manuscript.

 Answer) Thank you for your time to review our manuscript. We very appreciate your valuable comments and add our answers as point by point manner.

(Reviewer’s comment 2) The authors report “ The purpose of this study was to review the current use of animal-venom pharmacopuncture medicines in KM institutions, how such medicines are prepared, and for whatdisease treatment they are being used. Further, to provide basic data for the progressive, knowledge-based development of animal venom treatments derived from the understanding of the current status”.  However, in the present format, this review did not achieve these objectives. The authors should be more details about the toxic effect of these crude venom.

Answer) According to your comment, we added this point in the Discussion part as follows (pg 11 and 12).

  • While bee venom causes neurotoxic symptoms including local paresthesias, headache, dizziness, nausea and vomiting, muscle aches, and rarely, cerebrovascular infarcts [65, 66, 67, 68], recent non-clinical studies demonstrated that bee venom has anti-inflammatory [69], anti-nociceptive [70], and anti-cancer activities [71].
  • While snake venoms can be cardiotoxic (e.g. arrhythmias, bradycardia, tachycardia and hypotension) [77], myotoxic (e.g. myalgias, myopathy and rhabdomyolysis) [77], neurotoxic (e.g. ptosis, external ophthalmoplegia, dysphagia, dysphonia and broken neck sign) [78], and nephrotoxic (e.g. ischaemia and renal failure) [77], non-clinical studies of snake venom reported effects such as a decrease in blood pressure [79], regulation of blood coagulation [80], and anti-tumor [81] and analgesic activities [81].
  • While toad venom has toxic effects and cause gastrointestinal, mental, cardiac conduction, and arrhythmic disturbances [83], non-clinical studies have reported anti-inflammatory [84], antiplasmodial activity [85], and antiproliferative effect on cancer cells [86].

(Reviewer’s comment 3) Moreover, they could better describe the process of Preparation status of pharmacopuncture, especially the venom concentrations.

Answer) According to your comment, we added this point in the Results part as follows (pg 5). And, the clinical data were summarized by adding the contents related to venom concentration in table 2.

  • Bee venom is extracted from the venom pouch of worker bees (Apis mellifera, ligustica) [18]. Apply a week current of a low voltage, 3-6V, to a living bee. Then, take the venom coming out of the stinger [18]. Snake venom from the venom gland of Agkistrodon halys Pallas [18]. Hold a snake by its head and neck. Then, place a vessel under its fangs and press the head where the venom gland is located gently to get the venom out of it. This process is called ‘snake milking.’ [18, 19]. Toad (Bufo bufo gargarizans Canto) venom from the skin and parotid glands [18]. The production process of the injectable pharmacopuncture is the same for all three venoms (Figure 2). Venoms are processed, refined, and freeze-dried in a clean room into a powder formulation [18]. The venom powder goes through purification and dilution using distilled water to a specific concentration [18]. The venom is then filled, sealed, sterilized, and packaged as pharmacopuncture injections for clinical use (Fig. 2).

(Reviewer’s comment 4) The Questionnaire used in the study must be included in the review.

Answer) According to your comment, we added the Questionnaire as a supplementary material (Additional file1).

(Reviewer’s comment 5) In addition, reading the article I missed a more detailed table with the description of all the data obtained, highlighting the results of the therapy. Finally, the discussion can be improved based on clinical data.

Answer) Following your comment, we completed table 2 by adding details of clinical data including intervention (concetntraion, treatment sessions, amount of venom use) and adverse events. In addition, the following content has been added to the Discussion part (pg 12).

  • As is the case with bee stings, animal venom was used as it was, in some cases. However, in most cases, it was used as diluted to an appropriate concentration for clinical purposes. Animal venoms may cause idiosyncratic reactions depending on the individual in addition to their toxic effect [88]. For this reason, it is recommended to have the patient take a certain amount before clinical applications, in order to determine whether the use or administration of such a substance causes adverse effects or not. Also, according to the analysis on the 1,192,667 cases of accidents caused by animal venom over 12 years in Brazil by Chippaux [89], snake bites were linked to the highest fatality rate of 0.43%, followed by bee stings, of which the fatality rate was 0.33%. Since animal venom has toxic effects, it is necessary to refine, dilute, or otherwise process it before clinical use

Reviewer 2 Report

This paper reviews the use of animal venom pharmacopuncture in Korean traditional medicine based on the questionnaire administered by the Ministry of Health and Welfare and a literature review of scientific papers published in Korea. Animal venom-based medicines are prepared in external herbal dispensaries under strict control. Three kinds of venom, bee venom, snake venom, and toad venom, are used to treat musculoskeletal pain, control hemostasis, and treat cancer.

Comments for the authors:

  1. Please revise this sentence:

Bee venom therapy is effective in treating musculoskeletal pain, snake venom therapy in hemorrhage control in surgery, and toad venom therapy clinical effectiveness in cancer.

  1. The authors state that snake venom was obtained from the venom gland of Agkistrodon halys Pallas. Was it obtained by so-called »snake milking«?

  1. In the manuscript, there is Table 2 and Table 3, but I do not see Table 1.

Please check the bee venom data in Table 2. Is 99.306% the correct number?

In Table 3, in the column ‘Clinical evidence based on systematic review’, the citation numbers for bee venom should be checked.

  1. Figure 2.

Please revise the title of Figure A as follows:

Filtration of impurities and dilution with distilled water.

  1. Additional file 1.

As indicated in Section 3.1, the authors calculated the average number of pharmacopuncture medicines produced in the 11 P-EHDs, which is 18.4. The word "average" should be added to the table.

Author Response

Reviewer2

(Reviewer’s comment 1) This paper reviews the use of animal venom pharmacopuncture in Korean traditional medicine based on the questionnaire administered by the Ministry of Health and Welfare and a literature review of scientific papers published in Korea. Animal venom-based medicines are prepared in external herbal dispensaries under strict control. Three kinds of venom, bee venom, snake venom, and toad venom, are used to treat musculoskeletal pain, control hemostasis, and treat cancer.

 Answer) We appreciate your comment. We very appreciate your valuable comments and add our answers as point by point manner.

(Reviewer’s comment 2) Please revise this sentence: Bee venom therapy is effective in treating musculoskeletal pain, snake venom therapy in hemorrhage control in surgery, and toad venom therapy clinical effectiveness in cancer.

Answer) According to your comment, we revised the sentence in Abstract part (pg 1).

  • Bee venom therapy is effective in treating musculoskeletal pain, snake venom therapy is effective in controlling bleeding during surgery, and toad venom therapy is effective in cancer treatment

 (Reviewer’s comment 3) The authors state that snake venom was obtained from the venom gland of Agkistrodon halys Pallas. Was it obtained by so-called »snake milking«?

Answer) Thank you for your comment. In order to clarify the meaning, we supplemented the Results part (pg 5) as follows.

  • Snake venom from the venom gland of Agkistrodon halys Pallas [18]. Hold a snake by its head and neck. Then, place a vessel under its fangs and press the head where the venom gland is located gently to get the venom out of it. This process is called ‘snake milking.’ [18, 19].

(Reviewer’s comment 4) In the manuscript, there is Table 2 and Table 3, but I do not see Table 1.

Answer) Sorry for our mistake. We changed Table 2 and Table 3 into Table 1 (pg 4) and Table 2 (pg 7) respectively.

(Reviewer’s comment 5) Please check the bee venom data in Table 2. Is 99.306% the correct number?

Answer) According to your comment, we revised the number of table 2 (pg 4).

* Table 2 changed to Table 1.

(Reviewer’s comment 6) In Table 3, in the column ‘Clinical evidence based on systematic review’, the citation numbers for bee venom should be checked.

Answer) According to your comment, we have checked citation numbers for bee venom and revised the number of table 2 (pg 7).

* Table 3 changed to Table 2.

(Reviewer’s comment 7) Figure 2. Please revise the title of Figure A as follows: Filtration of impurities and dilution with distilled water.

 Answer) According to your comment, we revised the title of Figure 2-A (pg 6).

(Reviewer’s comment 8) Additional file 1. As indicated in Section 3.1, the authors calculated the average number of pharmacopuncture medicines produced in the 11 P-EHDs, which is 18.4. The word "average" should be added to the table.

Answer) According to your comment, we added the word “average” in Additional file 2.

* Additional file 1 changed to Additional file 2.

Round 2

Reviewer 1 Report

The authors have suscessfully addressed all my comments and suggestions. The changes significantly improved the work, therefore I endorse the publication of the study.